# Mexican Oregano (*Lippia graveolens* Kunth) as Source of Bioactive Compounds: A Review

**DOI:** 10.3390/molecules26175156

**Published:** 2021-08-25

**Authors:** Israel Bautista-Hernández, Cristóbal N. Aguilar, Guillermo C. G. Martínez-Ávila, Cristian Torres-León, Anna Ilina, Adriana C. Flores-Gallegos, Deepak Kumar Verma, Mónica L. Chávez-González

**Affiliations:** 1Bioprocesses and Bioproducts Research Group, Food Research Department, School of Chemistry, Universidad Autónoma de Coahuila, Saltillo 25280, Mexico; hernandez_israel@uadec.edu.mx (I.B.-H.); annailina@uadec.edu.mx (A.I.); or adrianaflores@uadec.edu.mx (A.C.F.-G.); 2Laboratory of Chemistry and Biochemistry, School of Agronomy, Universidad Autónoma de Nuevo León, General Escobedo, Monterrey 66050, Mexico; guillermo.martinezavl@uanl.edu.mx; 3Ethnobiological Garden and Research Center-UadeC (CIJE), Universidad Autónoma de Coahuila, Saltillo 27480, Mexico; ctorresleon@uadec.edu.mx; 4Agricultural and Food Engineering Department, Indian Institute of Technology Kharagpur, Kharagpur 721 302, India; deepak.verma@agfe.iitkgp.ernet.in

**Keywords:** *Lippia graveolens*, bioactive compounds, phenolic compounds, extraction techniques

## Abstract

*Lippia graveolens* is a traditional crop and a rich source of bioactive compounds with various properties (e.g., antioxidant, anti-inflammatory, antifungal, UV defense, anti-glycemic, and cytotoxicity) that is primarily cultivated for essential oil recovery. The isolated bioactive compounds could be useful as additives in the functional food, nutraceuticals, cosmetics, and pharmaceutical industries. Carvacrol, thymol, β-caryophyllene, and p-cymene are terpene compounds contained in oregano essential oil (OEO); flavonoids such as quercetin O-hexoside, pinocembrin, and galangin are flavonoids found in oregano extracts. Furthermore, thermoresistant compounds that remain in the plant matrix following a thermal process can be priced in terms of the circular economy. By using better and more selective extraction conditions, the bioactive compounds present in Mexican oregano can be studied as potential inhibitors of COVID-19. Also, research on extraction technologies should continue to ensure a higher quality of bioactive compounds while preventing an undesired chemical shift (e.g., hydrolysis). The oregano fractions can be used in the food, health, and agricultural industries.

## 1. Introduction

Oregano includes many aromatic species, mostly from the *Lamiaceae* and *Verbenaceae* families, which form most of the world’s commercially traded culinary herbs, primarily used for flavoring and traditional medicine for a variety of diseases (e.g., bacterial infections, digestive disorders, inflammatory diseases, and others) [1,2]. Furthermore, according to Cheikhyoussef et al. [3], the oregano crop is divided into four distinct groups: Turkish oregano (*Origanum onites*), Spanish oregano (*Coridohymus capitatu*), Greek oregano (*Origanum vulgare*) and Mexican oregano (*Lippia graveolens*). The Greek oregano (*O. vulgare*) has been the most studied oregano species, with many studies elucidating the potential as antioxidant, antimicrobial, antifungal, anti-inflammatory, and skin defensive auxiliaries associated with its rich polyphenolic fraction [4,5]. Similarly, several studies have been explored the potential of *L. graveolens* and other plants known as Mexican oregano synonyms (*Poliomintha longiflora*, *Lippia berlandieri* and *Monarda Fistulosa var. Menthifolia)* as auxiliary in oxidative disorders and antimicrobial properties [6,7,8,9].

In addition, oregano has been investigated as a source of bioactive compounds such as phenolic compounds, which are secondary metabolites produced by plants with antioxidant properties and a possible function in disease prevention [7,8,9]. The demand for bioactive compounds has increased; for example, the essential oil (EO) market is expected to hit $3226.2 million by 2025 [10]. According to the Codex Alimentarius Commission [11], Mexico is the second largest source, primarily of *L. graveolens* Kunth, *L. berlandieri Schauer,* and, more recently, for the *P. longiflora* species. The Mexican commercialization sequence has some disadvantages for farmers because of intermediaries that market the product with industry without developing more complex economic strategies [12].

The oregano plant is primarily used for EO recovery; oregano essential oil (OEO) is made up of terpenes (monoterpenes and sesquiterpenes), which are responsible for the aroma and strong flavor [13]. The key components of OEO are thymol and carvacrol, which are found in aromatic plants such as thyme (*Thymus vulgaris*), epperwort (*Lepidium flavum*), black cumin (*Nigella sativa*), and others [14]. Oregano plants are also high in bioactive polyphenolic compounds. These molecules have been isolated from raw leaves as a by-product of EO recovery. Even after a thermal extraction procedure, these by-products are a rich source of bioactive compounds [15].

In this context, the main purpose of this review article is to provide an overview of the *L. graveolens* and Mexican oreganos extraction techniques used for fraction recovery, chemical composition, variation, bioactivity, cultivation aspects, and intellectual property.

## 2. Chemical Composition

The oregano plant contains an essential oil fraction that generates most of its economic value as a flavoring and additive for food products. Furthermore, fresh leaves and wasted plant material have been suggested as a bioactive fraction and alternative source of bioactive compounds such as polyphenols (Flavonoids) and terpenes (Figure 1) with possible industrial applications.

### 2.1. Essential Oils (EOs)

The key fraction of metabolites of interest in oregano plant consists of essential oils (EOs), which have high volatility at room temperature and pressure and are composed of aromatic compounds, among other metabolites (e.g., esters, alcohols, aldehydes, and hydrocarbons) [16,17]. The compounds found primarily in EOs are terpenes (monoterpenes and sesquiterpenes) that are responsible for the aroma of aromatic plants; they are members of the mevalonate-based pathway active in the cytosol and are not dependent on mevalonate 2-C-methyl-D-erythriol 4-phosphate. Monoterpenes (C_10_H_16_) are composed of a 10-carbon chain that can be alicyclic, monocyclic, or bicyclic and contain unsaturated hydrocarbons and/or functional classes (e.g., alcohols). Sesquiterpenes (C_15_H_24_) are a more complex group of terpenoids since they can have a lactone ring and come in several forms such as linear, monocyclic, bicyclic, and tricyclic [18]. The composition and concentration of the metabolites found in EO vary according to the species studied. In the case of oregano, thymol and carvacrol are observed as main compounds, both with bioactive potential on human body and against industrially relevant microorganisms. Cid-Pérez et al. [19] identified the major chemical components (Table 1) present in EOs from Mexican oregano leaves (specifically *P. longiflora*), with thymol and carvacrol being the major compounds (1.97 ± 0.05 and 0.89 ± 0.10 mg mL^−1^, respectively). Similarly, previous studies using European varieties have revealed the presence of these compounds, with thymol having the highest proportion followed by carvacrol [20,21]. The EO extraction yield for oregano species varies, according to the literature 0.30–4.70%). This is dependent on the tissue, the culture conditions, the oregano species, and the extraction process (technology and conditions) [2,22,23,24].

Even though thymol and carvacrol are the most abundant compounds in *L. graveolens* essential oils, there are other compounds (Table 1) that are significant, such as *β*-caryophyllene, *p*-cymene, α-humelene, caryophyllene oxide, 1,8-Cineole, γ-Terpinene, and others [15,27,28,29]. The terpene and cannabinoid compound *β*-caryophyllene has significant anti-inflammatory potential and has been studied as a potential inhibitor of cellular proliferation (10 µg mL^−1^, 72 h) in glioblastoma through the interaction of CB2 and peroxisome proliferator-activated receptor gamma (PPARg) receptors [30]. Similarly, caryophyllene oxide is a sesquiterpene derived from various EOs [31]. It has been linked to antifungal and inflammatory properties [32,33]. Chavan et al. [33] reported its anti-inflammatory potential using a carrageenan-induced paw edema model; after three hours of the application of caryophyllene oxide (12.5 mg k^−1^ body weight) the volume of paw edema was statistically lower (0.45 ± 0.028 mL, * *p* < 0.05) than the control (Aspirin, 100 mg k^−1^ bodyweight) with 0.65 ± 0.064 mL. Otherwise, *p*-cymene and α-humelene (α-caryophyllene) are monoterpenes and sesquiterpenes, respectively. The compounds have demonstrated varying bioactivity as an adjunct against inflammatory, cancer, and microbial ailments [34,35]. Oliveira et al. [36] also conducted an in vivo study that demonstrated *p*-cymene potential against gastric lesions. The data suggested that *p*-cymene and rosmarinic acid (50–200 mg kg^−1^) significantly decreased the ulcer area; the findings could be attributed to antioxidant and immunomodulatory properties.

Otherwise, the main components of oregano have been reported against medical relevant viruses, with carvacrol reportedly showing antiviral potential against HRSV (human respiratory syncytial virus) and RV (human rotavirus), the data reported that the chemical application (pre/post-viral inoculation) showed a EC_50_ value between 55.9 to 123 µg mL^−1^ [37]. Additionally, carvacrol and thymol were reportedly shown to inhibit HIV-1 replication (IC_50_ value: 16 ± 2.9 µM and 25.2 ± 4.9 µM), cholesterol depletion of viral membrane and a blocking viral fusion on the HIV-1 virus [38,39].

Finally, the bioactivities associated with oregano EO are mostly relevant to the health and food sectors, although there have been reports of applications as an auxiliary in agronomic aspects such as green pesticides. The most important examples of EO application can be found in a study conducted by Cui et al. [8], in which the OEO was tested against Methicillin-Resistant *S. aureus* or MRSA, and studies conducted in 2017 have shown promising results in the application of oregano EO (*O. vulgare*) in combinations with other oils (*Syzygium aromaticum* “cloves” and *Leptospermum scoparium* “manuka”) as a larvicidal agent against *Aedes aegypti* [40].

### 2.2. Polyphenolic Compounds (PCs)

Peoples are interested in oregano plants because of their possible use as auxiliaries in a variety of applications. Polyphenolic compounds (PCs) are the second most common category of compounds in nature, only second to cellulose, which includes a wide range of compounds with at least one aromatic unit or phenol and one group hydroxyl, with a general classification corresponding to flavonoids and non-flavonoids [41]. PCs in plants perform protective functions against both abiotic (UV radiation) and biotic (pathogens, herbivores, and insects), which has led to research into the potential of compounds as adjunct agents in oxidative processes involved in the health and food sectors, as well as antimicrobial activities against strains of interest [42,43,44]. Flavonoids are the PCs with the highest bioactivity and are thought to be health boosters and preventive supplements with high nutritional and therapeutic value. This group has a phenyl benzopyrone skeleton, which is made up of two phenyl rings (A and B) connected by a pyran ring. Similarly, the methylation and hydroxylation found in rings A and B vary between the different flavonoid families. Flavones, isoflavones, flavonols, and anthocyanins are examples of flavonoid compounds [41]. Non-flavonoids, on the other hand, are compounds with a diverse structure that can include complex structures with high molecular weight. The group’s main representative is phenolic acids, which are composed of a phenyl group, a carboxylic functional group, and one or more hydroxyl. These compounds are classified as hydroxybenzoic acids (C_6_–C_1_) and hydroxycinnamic acids (C_6_-C_3_), with the former corresponding to an aromatic ring, a carboxylic group and one or more hydroxyl, while the latter indicates the existence of an additional chain with a double bond (C = C) and the carboxylic group [41,42].

The bioactivities related to oregano plant material extracts could be the result of the presence of phenolic compounds (flavonoids and phenolic acids) [44,45,46]. Cid-Pérez et al. [19] discovered the presence of phenolic acids (caffeic acid and rosmaric acid) *P. longiflora* residues. Similarly, compilation works (Table 1) of the most recent studies on oregano species have highlighted the existence of flavonoids and phenolic acids primarily glycosylated in plant tissues, such as luteolin, kaempferol, quercetin, eriodyctiol, naringenin, caffeic acid, gallic acid, and others [4,15,25,26].

A relevant bioactive compound present in *L. graveolens* extracts such a galangin has been shown to have antimicrobial, antimutagenic, antioxidant, anti-inflammatory, skin protector, and antiproliferative potential [47]. Hesperidin, another important compound, has been shown to have a possible auxiliary effect against fibrosis and hepatic oxidative stress by increasing hepatic antioxidant response [48]. Similarly, the use of naringenin potential against hepatotoxicity has been studied, and the results indicate a reduction in lipid oxidation, stabilization of the antioxidant response, and tissue protection [49]. The lithospermic acid and lithospermic acid B isolated from *Salvia miltiorrhiza* were found to be efficient and selective integrase inhibitors (IIs) on in vitro antiviral studies [50]. Otherwise, phenolic compounds, such as sakuranetin, present in Mexican oregano species have been previously isolated from *Baccharis retusa DC*, and it showed antileishmanial potential against protozoa of the genus *Leishmania* (*L. amazonensis* and *L. braziliensis*) with IC_50_ values of 51.89 to 45.12 μg mL^−1^, respectively [51]. Similarly, given the material’s richness, the study of the residual material of the essential oil fraction is important, as the material is rich in other compounds of interest that can promote product added value.

### 2.3. Oregano by-Products a Source of Polyphenolic Compounds

The extraction techniques are primarily concerned with the acquisition of targeted molecules (EOs), which constitute most industry’s economic profits. Otherwise, the residual material is discarded, a by-product problem occurs because of the large residual volume caused by the lower EOs extraction percentage (1–3%), the high generation of food waste presents potentially serious contamination problems, and high handling costs [52]. Mendez-Tovar et al. [53] identified that using by-products for energy generation or composting is a popular alternative, but the residual compounds may have antibacterial properties.

Furthermore, the residual material can be a rich source of bioactive compounds [15,19]. The by-products of *L. graveolens* are a rich source of bioactive compounds that could be valorized to increase the economic income of Mexican farmers. Arias et al. [15] identified 13 flavonoid compounds in *L. graveolens* by-products, highlighting the existence of apigenin, pinocembrin, galangin, and other compounds with well-studied bioactivities. Similarly, the value of *O. vulgare* ssp. *hirtum* residual material was assessed, and the key components of the hydro-alcoholic extract were rosmarinic acid, lithospermic acid, and glycosides flavonoids [54]. The residual extracts could be labeled as by-products; additionally, the residual extracts by microwave-assisted hydrodistillation (MAHD) revealed chlorogenic acid, caffeic acid, and luteolin-7-O-glucoside as the key components for *O vulgare*. Similarly, caffeic acid and rosmarinic acid have been found in Mexican oregano (*P. longiflora*) [19].

## 3. Extraction Techniques

The extraction process, which allows to recover a particular fraction that, after purification, can be applied to a food commodity (functional food market), is a critical step in exploiting the potential additional values of bioactive compounds [55,56]. Furthermore, the market increases the quest for improved extraction methods with lower operating costs and environmental friendliness by reducing organic solvent (hexane, chloroforms, ethyl acetate, etc.) in addition to a lower energy intake without compromising the consistency and quantity of the desired molecules (chemical structure stability) [57,58,59]. The following techniques (Figure 2) comprise current extraction procedures to recover secondary metabolites or those with potential application in Mexican oregano species.

### 3.1. Conventional

#### 3.1.1. Hydrodistillation

Hydrodistillation (HD) is a conventional and simple process for EO extraction that is used all over the world. Its applications include clove (*Syzygium aramaticum*), lavender (*Lavandula officinalis*), laurel (*Laurus nobilis*), oregano varieties, and other aromatic plants [60]. The plant material is immersed in boiling water as part of the technology (Figure 2). Following that, the emitted steam is directed to a condenser/separator to collect and isolate the EO from the water. Decantation is used in the separation process to increase the hydrophobicity of the EO. The HD process produces two by-products: residual water or aromatic waters “hydro-extract” and waste plant material. According to Cinbilgel et al. [61], residual water from EO processing was previously used in the subsequent extraction process, but it is now being sold as a commercial commodity by some oregano producers in Turkey. In addition, the reduced activity of HD-extracts from other plants for the bioreduction of Pd(II) to Pd(0) nanoparticles has been stated [62]. The application of HD in oregano crops mainly involves a long time extraction process (1–4 h) with large amounts of water; also, it is important to highlight that the extraction yield can be affected by steam flow [2]. The HD technology englobes an extraction yield range for Mexican oregano from 1.95 ± 0.15% to 2.26 ± 0.12% *v/w* [1]. Otherwise, in a recent year, the search of new “greener” ways to metabolites recovery on oregano species has proposed the Ohmic Heating-Assisted Hydrodistillation (OAH); the OAH comprises a lower extraction times with non-significance differences (*p* > 0.05) between extraction yield against HD with similar antioxidant activity [63].

The benefits of HD technology include a quick and easy-to-handle operation, as well as the capacity of water being immiscible with the essential oil fraction, which facilitates separation. Otherwise, the drawbacks include a long operating time (between 1 and 24 h), polar molecule losses, and chemical changes caused by interaction with boiling temperatures [64,65].

#### 3.1.2. Steam Distillation

Steam distillation (SD) refers to a group of methods that use steam as an extraction agent in hydrodistillation. The distinction between steam distillation and previous technologies is that there is no interaction between the plant material and water in steam distillation. SD may also be defined as direct or dry. In the direct phase, the target material is assisted in the same apparatus with water at the bottom that is heated to produce steam, and the volatile compounds are carried by the steam where they are diffused. Otherwise, in the dry phase, steam is produced in a separate (Figure 2) outside boiler and transported via plant material [66]. Some advantages include a solvent-free extraction, a low-cost operation, simple handling technology, and no subsequent separation steps. Furthermore, the distance between the steam generator and the target material has the benefit of minimizing EO variations [67]. The SD technology englobes an extraction yield range for Mexican oregano from 0.92% to 4.41% *v/w* [2,19]. In addition, the OEOs recovered by SD and HD have a similar chemical composition with thymol, carvacrol and cymene isomers as main components [1,2,19,63].

#### 3.1.3. Extraction with Organic Solvents

The process (Figure 2) employs organic solvents (chloroform, methanol, hexane, ethyl acetate) for compound extraction, allowing that the extraction takes place at lower temperatures and preventing the alterations (e.g., hydrolysis, deprotonations, etc.) caused by high temperatures compared to the previous methodologies. The drawback of its implementation is the residual contaminating material, which jeopardizes product protection [64,67]. Further, the organic solvents in Mexican oregano studies have been used to obtain a terpene fractionation (methanol, chloroform, and acetone) with an evaporation step [25]. The risk of organic solvent application was explored in the evaluation of *L. graveolens* organic extracts (hexane) on male CD-1 mice with a LD_50_: 1000 mg kg^−1^ (LD_50_ = Lethal Dose 50%) [68].

### 3.2. Emerging Technologies

#### 3.2.1. Ultrasound-Assisted Extraction (UAE)

Ultrasound-assisted extraction is a new technique that has significant advantages over its traditional predecessors, such as not requiring high operating temperatures, a lower degree of difficulty, no hazardous waste, high efficiency, and increased the quality of the targeted molecules [65,69,70]. Mohammadpour et al. [70] compared secondary metabolite extraction methods using Soxhlet (Conventional) against UAE in *Moringa peregrina*, and the results showed an improvement in the evaluated bioactivity (DPPH• and total phenols). The authors concluded that the UAE approach improves extraction efficiency. The technology (Figure 2) consists in the use of acoustic energy in conjunction with appropriate solvents to extract targeted molecules from plant matrix, the formation of bubbles due to cavitation phenomena, and their implosion results in a shear force that splits the cell, allowing compounds to be released into the extracting solvent [64,71]. The UAE has been used in oregano species for metabolite recovery; Oreopoulou et al. [54] reported a kinetic study of the phenolic compounds extraction process on oregano (*Origanum vulgare* ssp. *hirtum*) by-product of EO extraction process, the experimental data suggest a higher recovery of polyphenols using: 60% ethanol: water, particle size < 600 µm, temperature 22 °C and a solid: liquid ratio of 1:20 under the specifications of the extraction system used. *L. graveolens* extracts obtained by UAE have been applied on active packaging for blackberries with resveratrol, catechin and luteolin as identified compounds [72].

#### 3.2.2. Microwave-Assisted Extraction (MAE)

Microwave-assisted extraction is a novel technology that combines conventional solvent extraction with microwave (Figure 2) energy for bioactive compound extraction from plant matrix using magnetic and electrical fields with frequency ratios ranging from 300 mHZ to 300 GHz [73]. The energy system offers two mechanisms for molecular motion: ion migration and dipole rotation [74]. The technology involves a detachment between solute and active sites in the sample matrix under high energy conditions (temperature and pressure), followed by a solvent diffusion through the matrix and the matrix releasing the targeted compounds into the solvent [75,76]. Applications of MAE have advantages such as a low-temperature gradient (targeted heating), quicker curing times, better extraction yields, smaller facilities, and the ability to use “green” solvents instead of organic solvents (chloroforms, hexane, etc.). The microwave efficiency is highly dependent on solvent and matrix properties (dielectric constants) and combining a certain solvent with a plant matrix can increase extraction and procedure consistency. According to Llompart et al. [74], there is an unusual mechanism suitable for thermolabile compounds in which only the sample matrix is heated, and the solutes are released into the cold solvent. The MAE technology has been applied in other oregano species as *O. vulgare* for EO recovery by Microwave-assisted hydro-distillation (MAHD) with higher recovery yields (2.55%–7.10%) compared to 5.81% for HD [77].

#### 3.2.3. Supercritical Fluid Extraction (SCFE)

As an alternative to traditional methods, supercritical fluid extraction (SCFE) technology has been used in recent years for the extraction of bioactive compounds in complex matrices. As extraction agents, the SCFE used fluids at or above (Figure 2) supercritical temperatures and pressures [78,79,80,81,82]. Since supercritical fluids share properties with liquids and gases (density like liquids and viscosity like gases), they can penetrate deeper and faster into the matrix [83,84,85]. Supercritical CO_2_ is often used due to its chemical properties (critical point 7.38 MPa and 304.2 °K) as well as its safety (non-toxic). Furthermore, the solvent creates a non-oxidizing extraction environment. The benefits include a green and sustainable technology, lower energy requirements, low-temperature operation, higher extraction yields, strong selectivity, solvent recycling, and higher efficiency of targeted molecules [65,66,79,80,81,82,86,87]. Instead of molecules with low thermal stability that could be compromised by traditional thermal technology, the SCFE offers a safer alternative to bioactive compound extraction.

The general procedure consists of two stages: (i) extraction and (ii) extract isolation from the solvent. The supercritical fluid absorbs the solute in the first step (convection and diffusion). Following that, the pressure is decreased, as is the solvent strength, precipitating the solute (stage 2) [66,79,80,81,82,84,88]. To greatly improve the extraction performance of polar materials, a cosolvent (ethanol and methanol) can be used [83]. The SCFE application has been studied on *L. graveolens* by-products for flavonoid recovery using an optimization factorial design with extraction variables (pressure, temperature, flow, particle size, extraction time and ethanol percentage) [89].

### 3.3. Chemical Variation of Extraction Techniques

Because plant material and targeted compounds are subjected to various extraction conditions (EC), the choice of an extraction process has a significant effect on the chemical structure of the plant extract and the bioactivity relevant to phytochemical compounds (temperature, pressure, solvent, and others). Otherwise, the EC will have an impact on the density and solubility of the extraction fluid [89]. Similarly, other ECs such as solvent concentration, particle size, plant: solvent ratio, and temperature have been stated to have a significant impact on bioactive oregano extracts. For example, the extraction selectivity for polyphenolic compounds (%) with respect to temperature in oregano plants was higher before 40 °C (25.22 ± 0.42%) and decreased when the process reached 60 °C (19.36 ± 0.11%). According to the results, higher temperatures can cause thermal degradation of thermo-sensitive compounds [54]. Thus, the relationship between ECs and targeted material is critical in the recovery of bioactive compounds.

Several experiments have been conducted to determine the effect of extraction methods on plant extract bioactivity. García-Pérez et al. [90] investigated the statistical correlation and thermodynamics of supercritical-CO_2_ extraction in Mexican plants. The results indicated that the mixture of pressure and solvents influences antioxidant activity (expressed as radical DPPH^●^ scavenging and FRAP assay).

There is some evidence indicating that the concentration of oregano EOs’ key components could be responsible for bioactivity. Borgarello et al. [91] showed that residual fractions of OEO using molecular distillation had thymol concentrations up to 2.4 times greater than raw essential oil and higher antioxidant function. Similarly, a comparison of MAE and HD in OEO revealed a lower percentage of oxygenated monoterpenes (2.87–11.81%) in HD extracts, which might be due to thermal and hydrolytic conditions in which oxygenated monoterpenes convert to hydrocarbon monoterpenes [77]. High temperatures can degrade heat-labile phenolic compounds or cause polymerization under high pressures [46]. For example, Ameer et al. [92] confirmed that temperature decreased hesperidin yield on citrus peel for a longer extraction process with elevated temperatures. It is worth noting that its aglycone type has also been identified as a flavonoid present in Mexican oregano species [44].

Finally, differences in chemical compounds and bioactivity associated with extracts have been identified in other aromatic plants. For example, Conde-hernandez et al. [93] demonstrated that antioxidant activity of *Rosmarinus officinalis* EOs obtained by various technologies (SCFE, SD, and HD) showed a bioactivity difference, with supercritical extraction resulting in 14 times more antioxidant activity than steam distillation and hydrodistillation. As a result, an extraction optimization approach can also be used to build a better process to ensure the chemical stability for most bioactive compounds as well as a green extraction.

## 4. Bioactive Potential of Mexican Oregano Fractions

The recovery of secondary metabolite fractions of plant content from oregano has been used in evaluations for their activity as an auxiliary in antioxidative and antimicrobial methods, inhibition, dietary supplementation, and the production of green pesticides (Figure 3 and Table 2).

### 4.1. Food

The main use of oregano plants in the food industry is due to the organoleptic properties and flavoring properties in traditional foods. Recently, some investigations have explored the use of oregano fractions (EO or extracts) as a green auxiliary for food stability. For example, the antioxidant potential of *L. graveolens* extracts has been assessed (Table 2) using free radical scavenging methodologies such as DPPH• to elucidate the extracts’ ability as reactive oxygen species scavengers (ROS). The observed IC_50_ values vary from 21.89 ± 0.63 to 208.60 ± 12.25 μg mL^−1^, an interval that would be influenced by the fraction collected from the plant, since certain extractions performed favor the recovery of polar or non-polar compounds and report various bioactivities [19,94]. Similarly, *P. longiflora* plant classified as Mexican oregano was investigated by Lu-Martínez et al. [96] for *P. longiflora* EO fractions in *Prunus seronite var. capuli* (black cherry) oil as a natural source of antioxidants to have an extended life by slowing lipid oxidation. The findings indicated that a higher concentration (>3000 mg L^−1^) would induce pro-oxidation and a lower concentration (3 mg L^−1^) controls the formation of hydroperoxides. Similarly, the antioxidant ability of oregano as a lipid oxidation inhibitor has been investigated by its use in rich fatty acid beverages, with promising inhibition percentages (88–55%) [97].

Moreover, the oregano plant contains a high concentration of bioactive compounds (thymol, carvacrol, and caryophyllene oxide), which have been linked to antifungal/antibacterial properties and represent a green alternative for microorganism regulation of food-relevant strains. As an alternative, the use of OEOs could be a relevant approach, but the downside is that high volatilization can reduce process effectivity. The microemulsions should be used, and especially the oregano microemulsions demonstrated a lower MIC (Minimum Inhibitory Concentration) and MBC (Minimum Bactericide Concentration) by encapsulated EO of 2.25 and 4.5 (*v/v*%) for *Listeria monocytogenes* and *Escherichia coli*, respectively. Nonencapsulated EO, on the other hand, allowed a higher concentration of 9 (*v/v*%) [98].

Furthermore, food science in recent years has been focusing on the advancement of functional foods that have health benefits in addition to nutritional value. According to Gutierrez-Grijalva et al. [4], Mexican oregano (*L. graveolens*) can have significant potential as an auxiliary antioxidant and against enzymes involved in lipid and carbohydrate metabolism. However, their data indicate a difference depending on the digestion process, resulting in decreased activity for α-glucosidase inhibition and increased activity for α-amylase and lipase inhibition. The biomolecule heterogeneity in *L. graveolens* distinguishes the crop as a good source of bioactive compounds.

### 4.2. Health

Because of the rich bioactive fraction from plant extracts, the *L. graveolens* plant has been commonly used in traditional medicine as a strong auxiliary against certain ailments linked to microorganism infections or inflammatory processes. The main components of oregano extracts have been associated with the capability of bacterial control, including against antibiotic-resistant strains. Cui et al. [8] demonstrated the antibacterial activity of OEO against *S. aureus* (MRSA) and proposed a potential action mechanism that includes cell membrane disruption, negative respiratory chain interactions with main enzymes, carvacrol DNA interactions, and Panton Valentine leukocidin virulence factor (PVL) inhibition.

Furthermore, *P. longiflora* fractions (EO and hydroalcoholic extracts) have a MIC from 250 to 1000 mg L^−1^ against pathogenic bacteria such as *Staphylococcus aureus* and *Bacillus cereus* [19]. Furthermore, Reyes-Jurado et al. [13] evaluated the antimicrobial efficacy of essential oils (EOs) applied in the gaseous phase, the *Lippia berlandieri* assessment determined MIC values of 4 (*Escherichia coli)* and 0.28 (*Aspergillus niger)* μg mL^−1^ of air against relevant microorganism in the agricultural and food sectors. In addition, *L. graveolens* emulsion (Table 2) demonstrated an MLC_99_ (Minimum Lethal Concentration) range of 6.4 to 21.5 μL mL^−1^ emulsifier agent for *Candida albicans*, a relevant opportunistic human pathogen that causes candidiasis disease [28].

Furthermore, research on Mexican oregano extracts has been expanded to regulate ROS-related disorders in the body, taking advantage of the plant’s secondary metabolites with promising antioxidant potential. For *L. graveolens*, *L. palmeri,* and *Hedeoma patens* species, Leyva-López et al. [25] obtained positive findings in reducing the levels of ROS involved in inflammatory activities (59–87%), as well as inhibition of up to 78.2% and 81.7% of cyclooxygenases 1 and 2, respectively. The cyclooxygenases 2 inhibition assay measuring by PGF2α (ng/mL) production applied as positive control DuP-607 (DUP, 50 nM), DUP showed a statistically lower PGF2α concentration than oregano extracts. Furthermore, ethanolic extracts of *O. vulgare* are high in bioactive compounds such as rosmarinic acid, quercetin, apigenin, and carvacrol. Such bioactive compounds are also found in *L. graveolens* plants, which have been analyzed in terms of the inflammatory response by *Propionibacterium acnes*, demonstrating an inhibition of 32–37% of inflammation; the authors suggest that the obtained results could be the results of antimicrobial potential and a decrease in the related mRNA (messenger RNA) of interleukins involved in inflammatory processes such as IL-8 and IL-1β [99]. In another study, an analysis of secondary metabolite extraction, isolation and characterization revealed the bioactivity of the main flavonoids present in oregano extract, the higher inhibition of α-glucosidase (IC_50_ value of 37.199 μM) was obtained by Hispidulin. In addition, the anti-inflammatory potential was tested in an in vivo system and the activity (CD-1 mice) registered a IC_50_ range of 0.72 to 1.31 μmol/ear [97].

*L. graveolens* extracts have been evaluated as a possible protective agent against UV radiation (Table 2). The authors stated that using the extract decreased the lesions seen in laboratory rats exposed to UV radiation compared to the control group [94], and that it could have an anxiolytic effect in CD-1 mice based on experimental behavioral models. Accordingly, the crude extracts showed a similar effect as the reference drug diazepam (0.1 mg kg^−1^) [68]. Furthermore, it is also important to note that further research and information about anxiolytic properties in humans are needed.

### 4.3. Agronomic

Oregano residue is a rich source of bioactive molecules with possible health benefits in human and animal models. Several research studies have also centered on *O. vulgare* use as an ingredient for broiler feed, with possible benefits such as improved feed intake and conversion, disease regulation, improved digestion, fewer commercial antibiotics, and a lower economic effect for farmers [100]. Hernández-Coronado et al. [101] investigated the effects of two Mexican oregano species (*P. longiflora* and *L. berlandieri*) in drinking water on broiler meat quality. The authors discovered that OEO inclusion improved meat quality, and the sensory results revealed that treatment with *P. longiflora* produced the most suitable meat. Similarly, Bauer et al. [102] performed a study to determine the potential of *O. vulgare* as an antimicrobial agent applied to a diet (chickens) as a protective agent of the intestinal microbiota against pathogenic species such as *Proteus* spp., *Klebsiella* spp., and *Staphylococcus* spp., the study observed substantial inhibition at concentrations of 2% (*w/w*).

Furthermore, the use of EOs as green pesticides provides a relevant use due to the detrimental effects of conventional pesticides on ecosystem equilibrium, human and animal health, and chemical toxicity. The EO extracted from *L. origanoides* (a Mexican oregano) had an LC_50_ value of 53.79 mg L^−1^ against a health-relevant insect known as *Aedes aegypti (Diptera: Culicidae)*. In addition, the use of a combination of *L. origanoides* and *S. glutinosa* showed a potential synergistic effect with a lower LC_50_ (38.40 mg L^−1^) [103]. Similarly, the larvicidal effect of the main OEO components (carvacrol and terpinene-4-ol) against *Anopheles stephensi*, *A. subpictus*, *Culex quinquefasciatus* and *C. tritaeniorhynchus* gave LC_50_ of 21.15–27.95 μg mL^−1^ and 43.27–54.87 μg mL^−1^ [104]. In contrast, the EOs of *O. vulgare* ranged from 67 to 84.93 μg mL^−1^ [104].

Finally, Castilho et al. [105] investigated the agronomic tool of in vitro production for oregano plants, and the findings indicated that the addition of a plant growth regulator (Kinetin, 4.6 μM) yielded a greater number of compounds. Furthermore, the presence of indole-3-acetic acid (IAA) facilitates a high thymol and carvacrol content. As a result, agronomic in vitro production offers a modern method for higher quality plants with improved organoleptic and nutritional properties.

## 5. Research, Innovation, and Technological Perspectives

### 5.1. Intellectual Property

Several studies have evaluated the bioactive potential and chemical composition of Mexican oregano crops, but to elucidate the application of Mexican oregano in market products. World intellectual property databases such as Google Patents, a Google search engine that indexes patents, and PATENTSCOPE, a patent search tool from the World Intellectual Property Organization (WIPO), have been applied to search the technological growth in Mexican oregano beyond laboratory experiments.

According to the search made in Google Patents and WIPO-PATENTSCOPE (accessed March 29, 2021) using the keyword “*Lippia graveolens*”, 27 patents were registered in a period between 2004–2021. Of note, on Table 3 only seven patents use oregano in the formulations, two patents classified “oregano” as one of the main components in the formulation without a specific species. The patent ES2678597 is for a product made by a mixture of smokable herbs, which could be used as an auxiliary on smoking cessation, and it helps to clean and regenerate the lungs of the tobacco smoker [106]. The food technology patent (ES2613240) is focused on extending the shelf life of food by the creation of ice with antimicrobial activity by frozen drinking water and inclusion complexes formed by essential oils that are nano encapsulated with cyclodextrins. According to the technology, the melting ice will release bioactive compounds with antimicrobial activity [107].

Otherwise, four patents were directly related to the application of *L. graveolens* into the commercial product. The patent ES2351116 is comprised of an injectable solution containing isolated carvacrol and thymol of natural origin and anesthetic component to avoid discomfort; the application is via intramuscular or intravenous administration for use in human and veterinary medicine [108]. Similarly, the patent MXNL/A/2006/000057 belongs to health field in the application of *Larrea tridentata* extracts enriched with EOs from *L. graveolens* and medicinal plants with antimicrobial effects in humans and food models [109].

The following patents (Table 3) are related to the agronomic industry as biostimulants (ES2628278) that increase plant growth as a resistance inducer for the control of diseases caused by phytopathogenic viruses [110]. This includes an in vitro process (MX365079) for the regeneration of Mexican oregano (*Poliomintha genus*), which allows for sustainable agriculture schemes to be applied to this species, safeguarding the genetic material and the diversity [111]. Finally, the patent MX2017013652 describes a method to increase the secondary metabolites levels in oregano plants by abiotic stress. Thus, the technology opens perspectives with in vitro cultivars for the obtention of rich crops with potential application as a source of bioactive compounds for functional foods [112].

### 5.2. Post Pandemic Perspectives

For the past two years, humankind has been confronted with the massive spread of COVID-19, which has altered the everyday lives of millions of people [113]. Similarly, infection risk encourages market demand for functional foods and Food Complements as a preventive model, according to the literature [114,115]. As a result, the market relies on food products containing bioactive compounds to improve cellular activity, potentially lowering the prevalence and intensity of infection (bacterial, fungal, or viral) [116,117]. As a result, it is important to emphasize that incorporating bioactive compounds into a diet would not constitute a “cure,” and vaccines have been recommended as a response to COVID-19 by training our bodies to meet the infection.

Alternatively, some experiments show that bioactive substances can serve as auxiliaries against certain illnesses that can affect the proper functioning of the human system. Noor et al. [116] recently released a study demonstrating that the pandemic crisis triggers many psychiatric issues (paranoia, depression, insomnia, etc.) that can result in the production of oxidative stress, which can exacerbate the symptoms. Consequently, polyphenolic compounds (flavonoids, phenolic acid, and so on) have been suggested as health boosters due to their capacity to donate hydrogens or electrons to free radicals, thus stabilizing them [117]. The key component of OEOs, including carvacrol, demonstrated the ability to reduce lung tissue damage (edema, hyperemia, and lymphocyte infiltration) caused by influenza virus A, which has been identified because of an overreaction of the innate immune system [118].

Recently, Dr. Cristian Torres-León (co-author) and the CIJE research group used molecular docking (in silico) to assess the effectiveness of the most bioactive compounds found in *L. graveolens* against COVID-19 protease (Mpro) and replicase (RdRp) [119]. The findings revealed that Luteolin 7-*O*-glucoside has the maximum binding energy against Mpro, with a value of −8.2 kcal/mol. These findings outperformed those obtained with the antiviral chemical compounds lopinavir (−7.5 kcal/mol) and ribavirin (−6.4 kcal/mol). The binding energy of Luteolin 7-*O*-glucoside (−10.1 kcal/mol) against RdRp was found to be higher than that of the antiviral medications Remdesivir, Lopinavir, and Ribavirin (−8.4 to −9.9 kcal/mol). The findings show that the compounds found in Mexican oregano can serve as possible inhibitors of the COVID-19 Mpro and RdRp proteins. Finally, it can be suggested oregano bioactive compounds can be used to improve human health, but further research into kinetics, bioacceptability, biotransformation, and concentration is needed to determine the best scientific method.

## 6. Conclusions

The oregano plant comprises a crop with great economic relevance according to its commercialization as a culinary herb in the food industry; likewise, the phytochemical richness of the material allows it to be considered as a possible source of bioactive compounds with industrial potential (antimicrobial and antioxidant) of interest for the consumer. On the other hand, it is important to point out that technological development around Mexican species can be expanded through the incorporation of new extraction technologies and the development of new products that use the components present in the plant. Finally, it is important to note that current studies on Mexican oregano are mainly focused on the analysis of essential oils and hydroalcoholic extracts of fresh plant material, leaving the door open for future research that seeks to encourage the use of residual material resulting from EO extraction processes through characterization analysis, determination of bioactive potentials or favorable extraction methods that add an extra value to the plant.

## Figures and Tables

**Figure 1 molecules-26-05156-f001:**
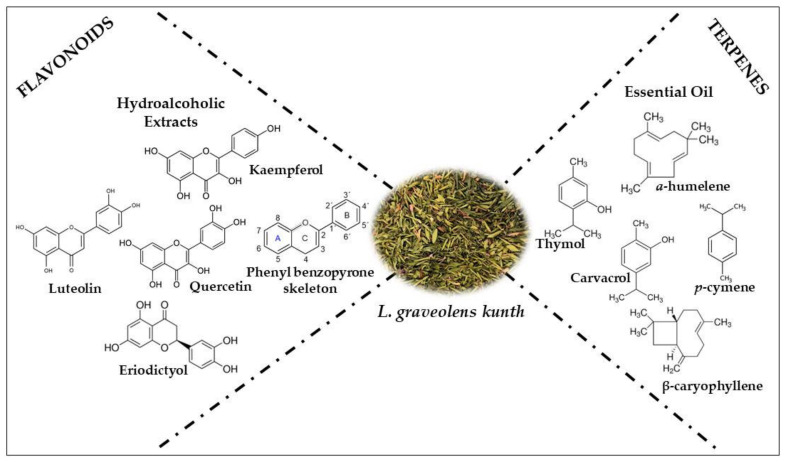
Chemical compounds present in *L. graveolens* Kunth *crops*.

**Figure 2 molecules-26-05156-f002:**
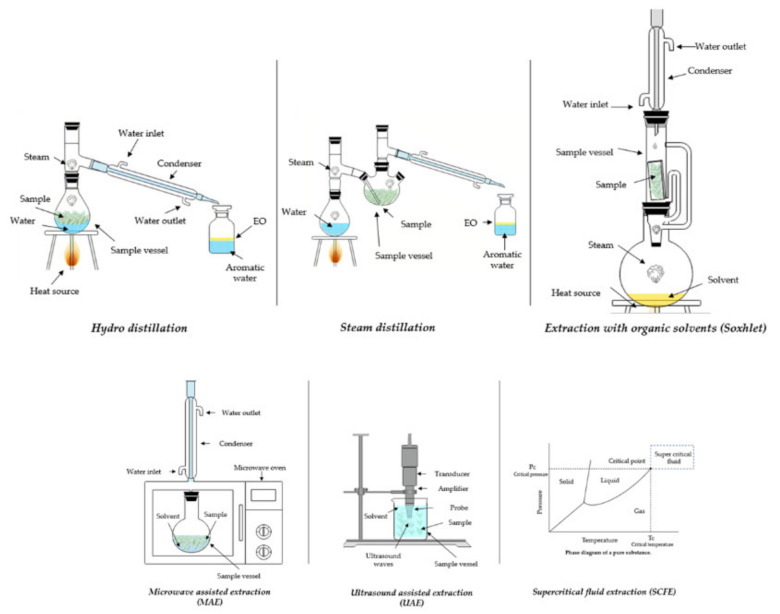
Extraction technologies applied in the recovery of secondary metabolites of Mexican oregano species [MAE: Microwave-assisted extraction; UAE: Ultrasound-assisted extraction; SCFE: Supercritical fluid extraction].

**Figure 3 molecules-26-05156-f003:**
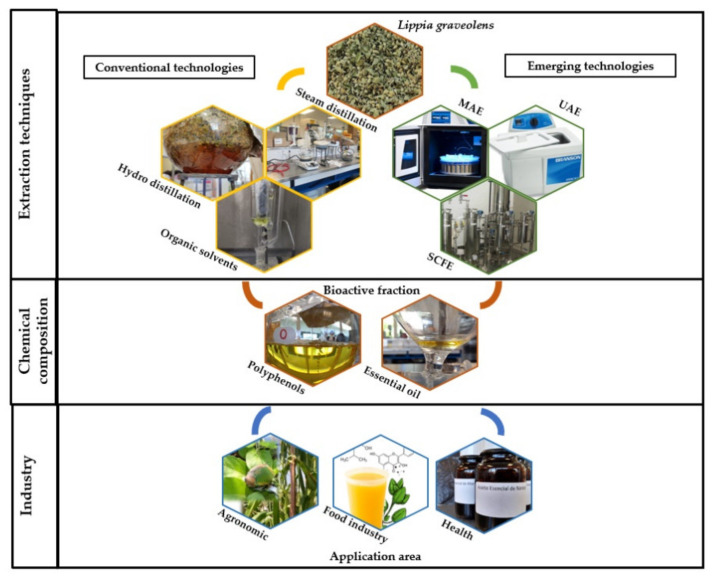
Extraction technologies, products, and application areas of Mexican oregano (*L. graveolens*). [MAE: Microwave-assisted extraction; UAE: Ultrasound-assisted extraction; SCFE: Supercritical fluid extraction].

**Table 1 molecules-26-05156-t001:** Main compounds identified in *L. graveolens* and other Mexican oregano species.

Fraction	Species	Type of Isolation	Mainly Identified Compounds	Content		Reference
				Concentration (mg mL^−1^/μg mg^−1^ Extract)	% EO	
Phenolic extracts	*L. graveolens*	Solid: liquid (80% methanol)	Quercetin O-hexoside/ Luteolin-glucuronide-glucoside/ Lithospermic acid/Pentahydroxy dihydrochalcone derivative	-	-	[4]
	*L. graveolens*	C0_2_ SCFE (80 bar 35 °C/ atmospheric 20 °C)	Eriodyctiol /Naringenin/ Sakuranetin/ Cirsimaritin/ Chrysoeriol	4.9 ± 0.2 ^B^/4.6 ± 0.2 ^B^/3.1 ± 0.29 ^B^/2.34 ± 0.04 ^B^/1.3 ± 0.1 ^B^	-	[15]
	*L. graveolens*	Solid: liquid (methanol/acetone/water) (50:40:10)	Quercetin-O-hexoside/ Scutellarein 7-O-hexoside/ Phloridzin/ Trihydroxy-methoxyflavone derivative/ 6-O-Methylscutellarein	-	-	[25]
	*L. graveolens*	Solid: liquid 1:20 (58% ethanol)	Pinocembrin/ Galangin/ phlorizin/ Naringenin/Quercetin/ Hispidulin/ Taxifolin	3.231 ± 0.390 ^A^/ 3.231 ± 0.390 ^A^/3.231 ± 0.390 ^A^/3.231 ± 0.390 ^A^/ 0.018 ± 0.008 ^A^/0.022 ± 0.003 ^A^/0.022 ± 0.003 ^A^		[26]
Essential oil	*L. berlandieri*	Commercial	*p*-cymene/Carvacrol/ *β*-pinene/Caryophyllene/ Camphene/ α-pinene	-	35.54/26.86/4.69/4.50/4.10/3.89	[13]
	*P. longiflora*	Clevenger-type apparatus	Carvacrol/ Thymol acetate/ Carvacrol, methyl ether/ Terpinolene/ *p*-cymene/ Borneol/ *β*-pinene	-	23.31/17.06/7.81/6.96/6.7/4.36/ 3.57	[19]
	*L. graveolens*	Hydro-distillation	Different chemotypes Carvacrol/ Thymol/ *β*-caryophyllene	-	53.05/70.6/27.6	[27]
	*L. graveolens*	Commercial	Thymol/ *p*-cymene/ Carvacrol/ *β*-caryophyllene/1,8-Cineole/ *γ*-terpinene	-	31.66/18.72/14.57/5.62/3.44/2.42	[28]

The letter means: A = mg mL^−1^ extract, B = μg mg^−1^ extract and % EO = Concentration on essential oil.

**Table 2 molecules-26-05156-t002:** Evaluations of the bioactive potential in Mexican oregano.

Specie	Analyzed Fraction	Extraction Technique	Tissue	Evaluated Bioactivity		Results	Reference
*L. berlandieri*	Essential oil	-	Leaves	Antimicrobial	*E. coli*/MRSA*/A. niger*	4/lower than 5/0.28 (MIC, μg mL^−1^ of air)	[13]
*P. longiflora*	Essential oil Et-OH extract Ethyl acetate extract	Hydrodistillation/Diffusion	Leaves	Antioxidant activity	DPPH^●^ *S. aureus*/*B. cereus*	83.70 ± 4.12 EO, 151.90 ± 6.65 E-OH, 208.60 ± 12.25 Et-Ac. (IC50, μg mL−1)	[19]
				Antimicrobial activity		250/250 EO, 1000/750 E-OH, 750/500 E-Ac (MIC, mg L−1)	
*L. graveolens* *L. palmeri*	Chloroform/ methanol extracts	Agitation/Sonication	Leaves	Antiflammatory	ROS reduction COX-1 and 2 cyclooxygenases inhibition	59.8% to 87% COX-1 78.2%/64.7%/67.8% COX-2 81.7%/74.6%/64.7%	[25]
*L. graveolens*	Essential oil	-	Leaves	Antimicrobial	*Candida albicans*	6.4 to 21.5 (MLC_99_, μL mL^−1^ emulsifier agent)	[28]
*L. graveolens*	Methanolic extract	Maceration	Aerial parts	Antioxidant/UV protection	DPPH^●^ *In vivo* penetration study	21.89 ± 0.63 (IC_50_, μg mL^−1^) 20.14 ± 1.86 (μg cm^−2^)	[94]
*L. graveolens*	Methanol extract	Percolation	Leaves/flowers	Antiglycemic	*α*-glucosidase inhibition	IC_50_ = 37.19 μM (Hispidulin).	[95]
				Anti-inflammatory	Antiflammatory	IC_50_ = 0.72–1.31 μmol/ear (Naringenin, Eriodictyol and 3-Hydroxyphloridzin).	
				Cytotoxicity	U251 & SK-LU-1 human tumor cell lines.	U251 (IC_50_ = 37.0 µM) SK-LU-1 (IC_50_ = 37.5 µM)	

EO = Essential oil, E-OH = Ethanol extract, E-Ac = Ethyl acetate extract, MIC = Minimum Inhibitory Concentration, MLC = Minimum Lethal Concentration, ROS = Reactive oxygen species, COX-1 = cyclooxygenase–1, COX-2 = cyclooxygenase–2, U-251 MG cell line human, SK-LU-1: Human Lung Cancer Cell Line.

**Table 3 molecules-26-05156-t003:** Reported patents in Google Patents and WIPO-PATENTSCOPE databases related with industrial application of oregano and Mexican oregano (*L. graveolens*).

Patent Number	Title	Main Core	Scope	Publication Data	Country	References
ES2678597	Smokable remedial herb blend	The present invention refers to a mixture of smokable herbs, which are part of a method of smoking cessation and help to clean and regenerate the lung of the tobacco smoker.	Health	14 August 2018	Spain	[106]
ES2613240	Composition of ice with antimicrobial activity, manufacturing method, and its applications.	The invention comprises a solution of frozen drinking water and inclusion complexes formed by essential oils nano encapsulated with cyclodextrins.	Food technology	23 May 2017	Spain	[107]
ES2351116	Antimicrobial therapeutic compositions and procedures for use.	The present invention relates to an injectable solution containing isolated carvacrol and thymol of natural origin for intramuscular or intravenous administration.	Health	31 January 2011	Spain	[108]
MXNL/A/ 2006/000057	Natural compounds having antimicrobial activity for preventing and controlling infectious diseases in humans and food.	The present invention refers to the use of water-soluble extracts in different concentrations of *Larrea tridentata* added with other natural products such as extracts and essential oils of the leaves of *L. graveolens* and other plants known to have an antimicrobial effect	Health	15 August 2006	Mexico	[109]
ES2628278	Biostimulant formulation of plant growth and development and inducer of resistance for the control of diseases caused by phytopathogenic viruses and method of preparation.	The biostimulant formulation is composed of extracts, vegetable oils from varieties of Chihuahuan semi-desert plants, absolute oils, and extracts from aromatic plants.	Agricultural biotechnology	2 August 2017	Spain	[110]
MX365079	Process for the regeneration of Mexican oregano plants (of the genus *Poliomintha*) by indirect organogenesis.	The invention refers to a process for the in vitro regeneration of Mexican oregano (*Poliomintha genus*), which allows a complete plant to be obtained from an explant.	Agricultural biotechnology	28 February 2014	Mexico	[111]
MX2017013652	Method to increase the secondary metabolites in candelilla and oregano.	The present invention relates to a method for increasing the secondary metabolites in crops such as candelilla and oregano by abiotic stressing.	Agronomic	24 October 2019	Mexico	[112]

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
