# Peer review of "Mexican Oregano (Lippia graveolens Kunth) as Source of Bioactive Compounds: A Review"

_molecules, 2021, doi:10.3390/molecules26175156_

Round 1

Reviewer 1 Report

My comments on the manuscript are as follows:

  1. Please consider moving topic 5.3 to the introduction.
  2. Line 46: “….has evolved [6]. As a source….. “Please correct the expression, something is missing.
  3. I think the authors should delete several words which could be repetitive. For example

Line 51: In this context…

Line 58: In this context…

  1. Lines 136-137: “The existence of phenolic compounds is due to the bioactivities present in various extracts of oregano plant material (flavonoids and phenolic acids)”. Please change the expression.
  2. In the section “3. Extraction techniques” the authors provide several techniques. Please explain if all these techniques are used in the case of Mexican oregano. Mention also any advantages or disadvantages, with the main goal always being the Mexican oregano.
  3. Line 297: “The main use of oregano plants in the food industry was due to the organoleptic properties”. Was ? not anymore?
  4. Lines 298-299: “Recent research has also focused at the use of oregano fractions (EO or extracts) as a green substitute for food stability”. Please change the expression.
  5. Lines 327-328: “As a result, the chemical structure of food as it passes through the digestive system is an important factor in its growth”. It doesn’t make sense, please change the expression.
  6. Lines 339-340: “The active ingredients in oregano extracts have been linked to bacterial dominance, including against antibiotic-resistant strains”. It doesn’t make sense, please change the expression.
  7. Lines 344: “Furthermore, P. longiflora has a MIC of 250 to 1000 mg L-1  against pathogenic bacteria”. The plant in general or a specific ingredient?
  8. Lines 345-346: “Furthermore, the Lippia berlandieri assessment determined MIC values of 4 and 0.28 μg mL-1 of air for Escherichia coli and Aspergillus niger”.  It doesn’t make sense, please change the expression.
  9. Line 365: “Other related bioactivities investigated in graveolens which had the ability as a UV radiation». It doesn’t make sense, please change the expression.
  10. Table 2:” Title: Evaluations of the bioactive potential in Mexican oregano”. ….. Please explain
  11. Lines 411-413: “According to the search made in "Google patents" and "WIPO-PATENTSCOPE" (accessed March 29, 2021) using the keyword "Lippia graveolens", 27 patents were registered in a period between 2004-2021”.  Please consider to add more information about the "Google patents" and  "WIPO-PATENTSCOPE", but no more than 2 to 3 lines.
  12. Lines 413-436: Given table 3, I think the text is unnecessary.
  13. The conclusions are very general and do not have the ability to give the reader a solid and concise point of view, about Mexican oregano as a source of bioactive compounds.
  14. Please discuss the methodological limitations of the article.

Author Response

  1. Please consider moving topic 5.3 to the introduction.

This suggestion was attended as suggested and now in the revised version we have included:

“ According to the Codex Alimentarius Commission [11], Mexico is the second largest source, primarily of L. graveolens kunth, L. berlandieri Schauer, and, more recently for P. longiflora species. The Mexican commercialization sequence has some disadvantages for farmers because of intermediaries that market the product with industry without developing more complex economic strategies [12].”   on lines 64-69.

  1. Line 46: “….has evolved [6]. As a source….. “Please correct the expression, something is missing.

This suggestion was attended as suggested and now in the revised version we have included:

“ Similarly, several studies have been explored the potential of L. graveolens and other plants known as Mexican oregano synonyms (Poliomintha longiflora, Lippia berlandieri and Monarda Fistulosa var. Menthifolia) as auxiliary in oxidative disorders and antimicrobial properties [6-9].”   on lines 54-58.

  1. I think the authors should delete several words which could be repetitive. For example

Line 51: In this context…

Line 58: In this context…

This suggestion was attended as suggested and now in the revised version we have included:

“ The oregano plant is primarily used for EO recovery…].”   on line 70.

  1. Lines 136-137: “The existence of phenolic compounds is due to the bioactivities present in various extracts of oregano plant material (flavonoids and phenolic acids)”. Please change the expression.

This suggestion was attended as suggested and now in the revised version we have included:

“ The bioactivities related to oregano plant material extracts could be the result of the presence of phenolic compounds (flavonoids and phenolic acids).”   on lines 198-200.

  1. In the section “3. Extraction techniques” the authors provide several techniques. Please explain if all these techniques are used in the case of Mexican oregano. Mention also any advantages or disadvantages, with the main goal always being the Mexican oregano.

This suggestion was attended as suggested and now in the revised version we have included more information related with the application of extraction technologies on Mexican oregano species:

“The application of HD in oregano crops mainly involves a long time extraction process (1 – 4 h) with large amounts of water; also, it is important highlight that the extraction yield can be affected by steam flow [2]. The HD technology englobes an extraction yield range for Mexican oregano from 1.95 ± 0.15 % to 2.26 ± 0.12 % v/w [1]. Otherwise, in a recent year the search of new “greener” ways to metabolites recovery on oregano species has propose the Ohmic Heating-Assisted Hydro distillation (OAH), the OAH comprises a lower extraction times with non-significance differences (p > 0.05) between extraction yield against HD with similar antioxidant activity [64]…” on lines 280 – 290.

“The SD technology englobes an extraction yield range for Mexican oregano from 0.92 % to 4.41 % v/w [2, 19]. Also, the OEOs recovered by SD and HD have a similar chemical composition with thymol, carvacrol and cymene isomers as main components [1, 2, 19, 64].” on lines 311 315.

“Further, the organic solvents in Mexican oregano studies have been used to obtain a terpene fractionation (methanol, chloroform, and acetone) with an evaporation step [45]. The risk of organic solvent application was explored in the evaluation of L. graveolens organic extracts (hexane) on male CD-1 mice with a LD50: 1000 mg kg-1 (LD50 = Lethal Dose 50%) [69].” on lines 323 -328.

“The UAE has been used in oregano species for metabolite recovery; Oreopoulou et al. [55] reported a kinetic study of the phenolic compounds extraction process on oregano (Origanum vulgare ssp. hirtum) by-product of EO extraction process, the experimental data suggest a higher recovery of polyphenols using: 60% ethanol: water, particle size < 600 µm, temperature 22°C and a solid: liquid ratio of 1:20 under the specifications of the extraction system used. Also, L. graveolens extracts obtained by UAE has been applied on active packaging for blackberry with resveratrol, catechin and luteolin as identified compounds [73]. “ on lines 344 – 354

“The MAE technology has been applied in other oregano species as O.vulgare for EO recovery by Microwave-assisted hydro-distillation (MAHD) with higher recovery yields (2.55 % – 7.10 %) compared to 5.81 % for HD [78].” on lives 379 -382.

“The SCFE application has been studied on L. graveolens by-products for flavonoid recover using an optimization factorial design with extraction variables (pressure, temperature, flow, particle size, extraction time and ethanol percent [90].” on lines 406 -409.

  1. Line 297: “The main use of oregano plants in the food industry was due to the organoleptic properties”. Was ? not anymore?

This suggestion was attended as suggested and now in the revised version we have included:

“ is due to the organoleptic properties and flavoring properties in traditional foods…”   on lines 464 - 465.

  1. Lines 298-299: “Recent research has also focused at the use of oregano fractions (EO or extracts) as a green substitute for food stability”. Please change the expression.

This suggestion was attended as suggested and now in the revised version we have included:

Recently, some investigations have explored the use of oregano fractions (EO or extracts) as a green auxiliary for food stability. “   on lines 466-467.

  1. Lines 327-328: “As a result, the chemical structure of food as it passes through the digestive system is an important factor in its growth”. It doesn’t make sense, please change the expression.

This suggestion was attended as suggested and now in the revised version we have included:

According to Gutierrez-Grijalva et al. [100], Mexican oregano (L. graveolens) can have significant potential as an antioxidant auxiliary and against enzymes involved in lipid and carbohydrate metabolism. “   on lines 503 – 506.

  1. Lines 339-340: “The active ingredients in oregano extracts have been linked to bacterial dominance, including against antibiotic-resistant strains”. It doesn’t make sense, please change the expression.

This suggestion was attended as suggested and now in the revised version we have included:

The main components of oregano extracts have been associated with the capability of bacterial control”  on lines 515-516.

  1. Lines 344: “Furthermore, P. longiflora has a MIC of 250 to 1000 mg L-1  against pathogenic bacteria”. The plant in general or a specific ingredient?

This suggestion was attended as suggested and now in the revised version we have included:  

“Furthermore, P. longiflora fractions (EO and hydroalcoholic extracts) have a MIC from 250 to 1000 mg L-1 against pathogenic bacteria such as Staphylococcus aureus and Bacillus cereus [19].” on lines 522-525.

  1. Lines 345-346: “Furthermore, the Lippia berlandieri assessment determined MIC values of 4 and 0.28 μg mL-1 of air for Escherichia coli and Aspergillus niger”.  It doesn’t make sense, please change the expression.

This suggestion was attended as suggested and now in the revised version we have included: 

“Furthermore, Reyes-Jurado et al. [13] evaluated the antimicrobial efficacy of essential oils (EOs) applied in gaseous phase, the Lippia berlandieri assessment determined MIC values of 4 (Escherichia coli) and 0.28 (Aspergillus niger) μg mL-1 of air against relevant microorganism in the agricultural and food sectors.”  on lines 525-529.

  1. Line 365: “Other related bioactivities investigated in graveolens which had the ability as a UV radiation». It doesn’t make sense, please change the expression.

This suggestion was attended as suggested and now in the revised version we have included:  “Also, L. graveolens extracts have been evaluated as a possible protective agent against UV radiation (Table 2)...”  on lines 560 - 561.

  1. Table 2:” Title: Evaluations of the bioactive potential in Mexican oregano”. ….. Please explain.

The table 2 comprise a summary of relevant studies with the objective to evaluate the bioactive potential of Mexican oregano fractions. As a result the title “Evaluations of the bioactive potential in Mexican oregano” englobes the different evaluation of Mexican oregano fractions.

  1. Lines 411-413: “According to the search made in "Google patents" and "WIPO-PATENTSCOPE" (accessed March 29, 2021) using the keyword "Lippia graveolens", 27 patents were registered in a period between 2004-2021”.  Please consider to add more information about the "Google patents" and  "WIPO-PATENTSCOPE", but no more than 2 to 3 lines.

This suggestion was attended as suggested and now in the revised we have included: 

“Several studies have evaluated the bioactive potential and chemical composition of Mexican oregano crops, but to elucidate the application of Mexican oregano in market products; world intellectual property databases as “Google patents” a Google search engine that indexes patents, and “PATENTSCOPE” a patent search tool from World Intellectual Property Organization (WIPO) have been applied to search the technological growth in Mexican oregano beyond laboratory experiments.”  on lines 610-617.

  1. Lines 413-436: Given table 3, I think the text is unnecessary.

  1. The conclusions are very general and do not have the ability to give the reader a solid and concise point of view, about Mexican oregano as a source of bioactive compounds.

This suggestion was attended as suggested and now in the revised we have included: 

“The oregano plant comprises a crop with great economic relevance according to its commercialization as a spice in the food industry, likewise, the phytochemical richness of the material allows it to be considered as a possible source of bioactive compounds with industrial potential (antimicrobial and antioxidant) of interest for the consumer. On the other hand, it is important to point out that technological development around Mexican species can be expanded through the incorporation of new extraction technologies and the development of new products that use the components present in the plant. Finally, it is important to note that current studies on Mexican oregano are mainly focused on the analysis of essential oils and hydroalcoholic extracts of fresh plant material, leaving the door open for future research that seeks to encourage the use of residual material resulting from EO extraction processes through characterization analysis, determination of bioactive potentials or favorable extraction methods that add an extra value to the plant. “ on lines 696 -711

Reviewer 2 Report

The manuscript deals with a topic of general interest, and does so by using a reasonable set of references and by treating a reasonable array of topics related to the proposed theme - Mexican oregano extracts. The topic is particularly challenging in ways often underappreciated - mostly due to the confusion brought about in general in the antioxidant community by vague references to health benefits that are supported by limited data but not by proper clinical studies. The following comments may be made, before an in-depth analysis of the manuscript is undertaken:

1. A key set of changes that would significantly benefit the manuscript would be to precisely identify the type of evidence behind the claimed health benefits. Most of these entail in vitro cell cultures, animal studies, or case studies - but not clinical trials or any other statistically significant human studies. As such, their "anti..." effects would arguably be better described as "potential anti..." until clear clinical or epidemiological analyses are available.

2. Where individual compounds are singled out for their health benefits, it would be useful to also state at what doses these effects occur, as well as stating how wide-spread the respective compounds are among other plants. 

3. When numerical data are quoted in the context of health benefits, such as MIC, it would be useful to also state how this compares to some "gold standards" - though, again, these are mostly in vitro measurements and it is therefore unknown how/if they would translate into clinical practice.

4. Some editing of the language is required. For instance, in Table 1 the term "mainly" should be replaced by "main". Also, the % column needs better explanation as it may cause confusion. Or, the term "auxiliary", which is used improperly in a few instances.

5. Figure 2 offers a nice synthetic view of the respective topic; it would fit in a talk/lecture - but it is too general and non-molecules-related to fit in a scientific journal paper at this level.

Author Response

Comments and Suggestions for Authors

The manuscript deals with a topic of general interest, and does so by using a reasonable set of references and by treating a reasonable array of topics related to the proposed theme - Mexican oregano extracts. The topic is particularly challenging in ways often underappreciated - mostly due to the confusion brought about in general in the antioxidant community by vague references to health benefits that are supported by limited data but not by proper clinical studies. The following comments may be made, before an in-depth analysis of the manuscript is undertaken:

  1. A key set of changes that would significantly benefit the manuscript would be to precisely identify the type of evidence behind the claimed health benefits. Most of these entail in vitro cell cultures, animal studies, or case studies - but not clinical trials or any other statistically significant human studies. As such, their "anti..." effects would arguably be better described as "potential anti..." until clear clinical or epidemiological analyses are available.

This suggestion was attended as suggested and now in the revised version we have changed the “anti…” effects for “potential”: 

“with many studies elucidating the potential as antioxidant, antimicrobial, antifungal, anti-inflammatory, and skin defensive auxiliaries…” on lines 51- 53

“β-caryophyllene has significant anti-inflammatory potential…” on line 124.

“to antifungal and inflammatory properties [31, 32]. …” on line 130

“that demonstrated p-cymene potential against gastric lesions,” on lines 140 -141.

“antimicrobial, antimutagenic, antioxidant, anti-inflammatory, skin protector, and antiproliferative potential …” on lines 208 – 209.

“shown to have a possible auxiliary effect against fibrosis and hepatic oxidative stress by increasing hepatic antioxidant response [49]. on lines 210-212.

“the use of naringenin potential against hepatotoxicity has been studied...” on line 212.

“], Mexican oregano (L. graveolens) can have significant potential as an antioxidant auxiliary and against enzymes involved in lipid and carbohydrate metabolism …” on lines 503 - 506.

“, the authors suggest that the obtained results could be the results of antimicrobial potential and a decrease in the related mRNA …” on lines 549 -551.

“the anti-inflammatory potential was tested on in-vivo system …” on lines 556 -557.

L. graveolens extracts have been evaluated as a possible protective agent against UV radiation.” on lines 559 -560.

  1. Where individual compounds are singled out for their health benefits, it would be useful to also state at what doses these effects occur, as well as stating how wide-spread the respective compounds are among other plants. 

This suggestion was attended as suggested and now in the revised version we have included:

β-caryophyllene has significant anti-inflammatory potential and has been studied as a potential inhibitor of cellular proliferation (10 µg mL-1, 72 h) in glioblastoma through the interaction of CB2 and peroxisome proliferator-activated receptor gamma (PPARg) receptors [29].” on lines 124 - 128.

“Chavan et al. [32] reported its anti-inflammatory potential using carrageenan-induced paw edema model, after three hours of the application of caryophyllene oxide (12.5 mg k-1 body weight) the volume of paw edema was statistically lower (0.45 ± 0.028 mL, *p < 0.05) than the control (Aspirin, 100 mg k-1 body weight) with 0.65 ± 0.064 mL.” on lines 131 - 136.

“Oliveira et al. [35] also conducted an in-vivo study that demonstrated p-cymene potential against gastric lesions, the data suggest that p-cymene and rosmarinic acid (50 -200 mg kg-1) significantly decreased the ulcer area, the findings could be attributed to antioxidant and immunomodulatory properties..” on lines 139 - 144.

“The lithospermic acid and lithospermic acid B isolated from Salvia miltiorrhiza were found to be efficient and selective integrase inhibitors (IIs) on in-vitro antiviral studies [51].” on lines 215 -218.

“Otherwise, phenolic compounds, such as sakuranetin present in mexican oregano species has been previosuly isolated from Baccharis retusa DC, and it has showed an antileishmanial potential against protozoa of the genus Leishmania (L. amazonensis and L. braziliensis) with IC50 values of 51.89 to 45.12 μg mL-1, respectively [52].” on lines 218 222.

  1. When numerical data are quoted in the context of health benefits, such as MIC, it would be useful to also state how this compares to some "gold standards" - though, again, these are mostly in vitro measurements and it is therefore unknown how/if they would translate into clinical practice.

This suggestion was attended as suggested and now in the revised version we have included:

“Chavan et al. [32] reported its anti-inflammatory potential using carrageenan-induced paw edema model, after three hours of the application of caryophyllene oxide (12.5 mg k-1 body weight) the volume of paw edema was statistically lower (0.45 ± 0.028 mL, *p < 0.05) than the control (Aspirin, 100 mg k-1 body weight) with 0.65 ± 0.064 mL.” on lines 131 - 136.

“The cyclooxygenases 2 inhibition assay measuring by PGF2α (ng/mL) production applied as positive control DuP-607 (DUP, 50 nM), DUP showed a statistical lower PGF2α concentration that oregano extracts.” on lines 541 -544

Note: The article does not report the specific value only graphic

“and that it could have an anxiolytic effect in CD-1 mice based on experimental behavioral models, the crude extracts showed a similar effect as the reference drug diazepam (0.1 mg kg-1) [69].” on lines 562 -565.

  1. Some editing of the language is required. For instance, in Table 1 the term "mainly" should be replaced by "main". Also, the % column needs better explanation as it may cause confusion. Or, the term "auxiliary", which is used improperly in a few instances.

This suggestion was attended as suggested and now in the revised version we have included:

“Main compounds identified…” on line 189.

“the larvicidal effect of the main OEO components…” on lines 595 -596.

“characterization revealed the bioactivity of the main flavonoids present in oregano extract” on lines 553 -554.

This suggestion was attended as suggested and now in the revised version we have included:

“%EO = Concentration on essential oil…” on line 190

  1. Figure 2 offers a nice synthetic view of the respective topic; it would fit in a talk/lecture - but it is too general and non-molecules-related to fit in a scientific journal paper at this level.   

    Modifications in figures were considered

Reviewer 3 Report

Article is total mix of unrelated, randomly selected chapters. Chaotic mix of many aspects written with many language errors. Suggested corrections:

  1. Chemical names should be unified.
  2. Line 106 - manuka is not EO.
  3. Figures have very poor quality, without legends.
  4. In "2. Chemical composition", Authors described on two pages building and rings of chemical compounds, but real compounds of. L. graveolens are presented in 9 lines (88-96). Rest of the text can be removed.
  5. Part "3. Extraction techniques" is without sense, and does not contribute anything that is related to the title of the article. No affiliation with bioactive compounds of Lippia graveolens.
  6. Only one, the most important part of article is "Table 2. Evaluations of the bioactive potential in Mexican oregano". But unfortunately, in this table are cited only 6 references. 
  7. I also do not understand why the authors put the chapter "5.1. Intellectual property", which is not related to the title of the article.
  8. Authors not described antiviral activity of L. graveolens compounds, and suddenly at the end of the article, it appeared "5.2. Post pandemic perspectives". So what if luteolin 7-O-glucoside has anti-SARS-CoV-2 activity in silico research? Has it been confirmed in vitro?
  9. And the next chapter, unrelated to anything else in this article, is "5.3. Economic aspect of Mexican farmers."

Unfortunately, article is very poor, illogical, without important details. Therefore, I do not recommend it for publication, even after corrections. The article would have to be completely rewritten.

Author Response

Article is total mix of unrelated, randomly selected chapters. Chaotic mix of many aspects written with many language errors. Suggested corrections:

  1. Chemical names should be unified.

This suggestion was attended as suggested and now in the revised version we have homogenized the chemical names:

β-pinene, p-cymene / Carvacrol/ β-pinene / Caryophyllene/ Camphene/ α-pinene Carvacrol/ Thymol/ β-caryophyllene Thymol/ p-cymene/ Carvacrol/ β-caryophyllene/1,8-Cineole/ γ-terpinene” on lines 189 -190.

  1. Line 106 - manuka is not EO.

According to Muturi et al., 2017 (DOI: doi: 10.1093/jme/tjx168)

“Twenty-one essential oils were initially screened for their toxicity against Aedes aegypti (L.) larvae and three out of the seven most toxic essential oils (Manuka, oregano, and clove bud essential oils) were examined for their chemical composition and combined toxicity against Ae. aegypti larvae.”

This suggestion was attended as suggested and now in the revised version we have included:

“studies conducted in 2017 have shown promising results in the application of oregano EO (O. vulgare) in combinations with other oils (Syzygium aromaticum “cloves” and Leptospermum scoparium “manuka”) as a larvicidal agent against Aedes aegypti [39]. on lines 159 -162.

  1. Figures have very poor quality, without legends.

This suggestion was attended as suggested and now in the revised version we have improved the image resolution.

  1. In "2. Chemical composition", Authors described on two pages building and rings of chemical compounds, but real compounds of. L. graveolens are presented in 9 lines (88-96). Rest of the text can be removed.

The section has been improved

  1. Part "3. Extraction techniques" is without sense, and does not contribute anything that is related to the title of the article. No affiliation with bioactive compounds of Lippia graveolens.

This suggestion was attended as suggested and now in the revised version we have included more information related with the application of extraction technologies on Mexican oregano species:

“The application of HD in oregano crops mainly involves a long time extraction process (1 – 4 h) with large amounts of water; also, it is important highlight that the extraction yield can be affected by steam flow [2]. The HD technology englobes an extraction yield range for Mexican oregano from 1.95 ± 0.15 % to 2.26 ± 0.12 % v/w [1]. Otherwise, in a recent year the search of new “greener” ways to metabolites recovery on oregano species has propose the Ohmic Heating-Assisted Hydro distillation (OAH), the OAH comprises a lower extraction times with non-significance differences (p > 0.05) between extraction yield against HD with similar antioxidant activity [64]…” on lines 280 – 290.

“The SD technology englobes an extraction yield range for Mexican oregano from 0.92 % to 4.41 % v/w [2, 19]. Also, the OEOs recovered by SD and HD have a similar chemical composition with thymol, carvacrol and cymene isomers as main components [1, 2, 19, 64].” on lines 311 315.

“Further, the organic solvents in Mexican oregano studies have been used to obtain a terpene fractionation (methanol, chloroform, and acetone) with an evaporation step [45]. The risk of organic solvent application was explored in the evaluation of L. graveolens organic extracts (hexane) on male CD-1 mice with a LD50: 1000 mg kg-1 (LD50 = Lethal Dose 50%) [69].” on lines 323 -328.

“The UAE has been used in oregano species for metabolite recovery; Oreopoulou et al. [55] reported a kinetic study of the phenolic compounds extraction process on oregano (Origanum vulgare ssp. hirtum) by-product of EO extraction process, the experimental data suggest a higher recovery of polyphenols using: 60% ethanol: water, particle size < 600 µm, temperature 22°C and a solid: liquid ratio of 1:20 under the specifications of the extraction system used. Also, L. graveolens extracts obtained by UAE has been applied on active packaging for blackberry with resveratrol, catechin and luteolin as identified compounds [73]. “ on lines 344 – 354

“The MAE technology has been applied in other oregano species as O.vulgare for EO recovery by Microwave-assisted hydro-distillation (MAHD) with higher recovery yields (2.55 % – 7.10 %) compared to 5.81 % for HD [78].” on lives 379 -382.

“The SCFE application has been studied on L. graveolens by-products for flavonoid recover using an optimization factorial design with extraction variables (pressure, temperature, flow, particle size, extraction time and ethanol percent [90].” on lines 406 -409.

  1. Only one, the most important part of article is "Table 2. Evaluations of the bioactive potential in Mexican oregano". But unfortunately, in this table are cited only 6 references. 

The potential bioactivities of this type of plants is well known, there is a lot of information regarding many varieties of oregano. In the case of Mexican oregano, few studies have been reported.

  1. I also do not understand why the authors put the chapter "5.1. Intellectual property", which is not related to the title of the article.

According to the bibliographic research, the authors realize that the "intellectual property" in oregano plants has not been studied, and we think that the publication of related information could be of interest to the reader, with the aim of making known the technological growth of Mexican oregano beyond laboratory experiments.

Also, it is important highlight that some patents (ES2628278, ES2351116, MXNL/A/2006/000057, ES2613240 and ES2678597) applied the potential bioactivity (antimicrobial and antioxidant) of oregano extracts or organoleptic properties (ES2678597) related to the chemical compounds. Also, the other patents making known that the oregano technological growth englobes agronomical topics as “in-vitro” cultivation technologies related with increase the crops production and metabolites.

  1. Authors not described antiviral activity of L. graveolens compounds, and suddenly at the end of the article, it appeared "5.2. Post pandemic perspectives".

This suggestion was attended as suggested and now in the revised version we have included information related with antiviral potential:

“Otherwise, the main components of oregano have been reported against medical relevant viruses, the carvacrol has reported antiviral potential against HRSV (human respiratory syncytial virus) and  RV (human rotavirus), the data reported that the chemical application (pre/post viral inoculation) showed a EC50 value between 55.9 to 123 µg mL-1 [36]. Additionally, carvacrol and thymol has reported a HIV-1 replication inhibition (IC50 value: 16 ± 2.9 µM and 25.2 ± 4.9 µM), cholesterol depletion of viral membrane and a blocking viral fusion on HIV-1 virus [37, 38]” on lines 144 -152.

“The lithospermic acid and lithospermic acid B isolated from Salvia miltiorrhiza were found to be efficient and selective integrase inhibitors (IIs) on in-vitro antiviral studies [51].” on lines 215-218.

So what if luteolin 7-O-glucoside has anti-SARS-CoV-2 activity in silico research? Has it been confirmed in vitro?

No, it has been. According to our bibliographical research the work made by Torres-León et al., 2020, it is the first publication of docking around Mexican oregano flavonoids against against COVID-19 protease (Mpro) and replicase (RdRp). However, the research focuses only on in-silico methodology that could be used as a screening work for future works focuses on in-vitro or in-vivo models. But we think that it is important highlight the recent work on oregano topics related with pandemic situation. 

  1. And the next chapter, unrelated to anything else in this article, is "5.3. Economic aspect of Mexican farmers."

This suggestion was attended as suggested and now in the revised version we have included the relevant information of the chapter on introduction:

“ According to the Codex Alimentarius Commission [11], Mexico is the second largest source, primarily of L. graveolens kunth, L. berlandieri Schauer, and, more recently for P. longiflora species. The Mexican commercialization sequence has some disadvantages for farmers because of intermediaries that market the product with industry without developing more complex economic strategies [12].”   on lines 64-69.

Unfortunately, article is very poor, illogical, without important details. Therefore, I do not recommend it for publication, even after corrections. The article would have to be completely rewritten.

Reviewer 4 Report

The authors reviewed Lippia graveolens Kunth

title suggestion

Mexican oregano Lippia graveolens Kunth (Verbenaceae) as a source of bioactive compounds - a review

  1. Extraction techniques

I lack feeling for figures that could illustrate the information contained in the text, as they can broaden the reader's understanding.

Extraction of supercritical fluids could be inserted into a phase diagram of a pure substance.

Add the references

  1. Silva, S.G.; de Oliveira, M.S.; Cruz, J.N.; da Costa, W.A.; da Silva, S.H.M.; Barreto Maia, A.A.; de Sousa, R.L.; Carvalho Junior, R.N.; de Aguiar Andrade, E.H. Supercritical CO2 extraction to obtain Lippia thymoides Mart. & Schauer (Verbenaceae) essential oil rich in thymol and evaluation of its antimicrobial activity. J. Supercrit. Fluids 2021, 168, 105064, doi:10.1016/j.supflu.2020.105064.
  2. Oliveira, M.S. de; Silva, S.G.; Cruz, J.N. da; Ortiz, E.; Costa, W.A. da; Bezerra, F.W.F.; Cunha, V.M.B.; Cordeiro, R.M.; Neto, A.M. de J.C.; Andrade, E.H. de A.; et al. Supercritical CO2 Application in Essential Oil Extraction. In Industrial Applications of Green Solvents – Volume II; Inamuddin, R.M., Asiri, A.M., Eds.; Materials Research Foundations: Millersville PA, USA, 2019; pp. 1–28, doi:10.21741/9781644900314-1.

Improve the quality of figure 2

Use of updated template from the Molecules (2021)

Author Response

The authors reviewed Lippia graveolens Kunth

title suggestion

Mexican oregano Lippia graveolens Kunth (Verbenaceae) as a source of bioactive compounds - a review

We consider keeping the title

  1. Extraction techniques

I lack feeling for figures that could illustrate the information contained in the text, as they can broaden the reader's understanding.

Extraction of supercritical fluids could be inserted into a phase diagram of a pure substance.

This suggestion was attended as suggested and now in the revised version we have included:

Figure 2. Extraction technologies applied in chemical compounds recovery in L. graveolens crops.  on lines 354 -358.

Add the references

  1. Silva, S.G.; de Oliveira, M.S.; Cruz, J.N.; da Costa, W.A.; da Silva, S.H.M.; Barreto Maia, A.A.; de Sousa, R.L.; Carvalho Junior, R.N.; de Aguiar Andrade, E.H. Supercritical CO2 extraction to obtain Lippia thymoides Mart. & Schauer (Verbenaceae) essential oil rich in thymol and evaluation of its antimicrobial activity. J. Supercrit. Fluids 2021, 168, 105064, doi:10.1016/j.supflu.2020.105064.
  2. Oliveira, M.S. de; Silva, S.G.; Cruz, J.N. da; Ortiz, E.; Costa, W.A. da; Bezerra, F.W.F.; Cunha, V.M.B.; Cordeiro, R.M.; Neto, A.M. de J.C.; Andrade, E.H. de A.; et al. Supercritical CO2 Application in Essential Oil Extraction. In Industrial Applications of Green Solvents – Volume II; Inamuddin, R.M., Asiri, A.M., Eds.; Materials Research Foundations: Millersville PA, USA, 2019; pp. 1–28, doi:10.21741/9781644900314-1.

This suggestion was attended as suggested and now in the revised version we have included the references:

Improve the quality of figure 2

 This suggestion was attended as suggested and now in the revised version we have improved the image resolution.

Use of updated template from the Molecules (2021)

This suggestion was attended as suggested:

Round 2

Reviewer 1 Report

Congratulations on your work. A minor observation is that the manustript needs  moderate english chages, for example line 70 ... is made up (consists mainly?).

Reviewer 2 Report

The authors have addressed my previous comments in reasonable manner.

Reviewer 3 Report

Authors corrected some points. Unfortunately, article all time is mix of unrelated, randomly selected chapters. If title of article is about "bioactive compounds", I as reader, would like read about antimicrobial, anticancer, antioxidant, anti-inflammatory, and other activities, but not about e.g. patents. If I want to read about patents, I will look for a decent article only on patents. Therefore, if title of reviewed article is about "bioactive compounds", it should be about ONLY "bioactive compounds".

Suggested corrections:

  1. Part "3. Extraction techniques" is without sense in relation to the title of the article and must be removed.
  2. I also do not understand why the authors put the chapter "5.1. Intellectual property", and "Post pandemic perspectives", which are not related to the title of the article.
  3. Only one, the most important part of article is "Table 2. Evaluations of the bioactive potential in Mexican oregano". But unfortunately, in this table are cited only 6 references. Please, search in PubMed and you can add about 20 references.

Unfortunately, all time article is very poor and chaotic. Therefore, I suggest reject it.